# LEARNING PROPORTIONAL ANALOGIES: LIGHTWEIGHT NEURAL NETWORK VS LLM

## ABSTRACT

Analogical reasoning often involves statements of the form $\alpha : \beta :: \gamma : \delta$, known as proportional analogies, which can be interpreted as "$\alpha$ differs from $\beta$ as $\gamma$ differs from $\delta$" and "$\beta$ differs from $\alpha$ as $\delta$ differs from $\gamma$". In this paper, we study the learnability of proportional analogies from both theoretical and experimental perspectives. We show that, in the Boolean setting—where each element of a proportional analogy is represented as a Boolean vector—proportional analogies are efficiently PAC learnable. To validate this in practice, we instantiate proportional analogies in a perceptual scenario with 4-cell images, each cell containing a shape and a color. We automatically generate a dataset of valid and invalid proportional analogies and train lightweight artificial neural networks (ANNs) as evaluators. We compare our ANN-based models against state-of-the-art Large Language Models (LLMs) in proportional analogy verification (checking correctness), proportional analogy generation (producing missing elements), and proportional analogy generalization (applying knowledge acquired during learning to unseen features). Our results show that lightweight ANNs i) match LLMs in verification and generalization, and ii) outperform LLMs in generation, demonstrating that simple, efficient models can effectively learn and generalize proportional analogies while using far fewer resources.

## 1 INTRODUCTION

Analogies were initially introduced in the works of Aristotle (2009). In recent years, they have garnered the attention of many researchers and have even been characterized as being "at the core of cognition" (Hofstadter, 2001; Gentner et al., 2001). Although such a strong stance might not be universally shared, analogies have sparked renewed interest over the last fifteen years, especially in NLP, with research on lexical analogies (Mikolov et al., 2013; Gladkova et al., 2016; Rogers et al., 2017), analogies between sentences (Zhu & de Melo, 2020), and collections of procedural texts (Sultan & Shahaf, 2022), as well as on the analogical reasoning capabilities of Large Language Models (LLMs) (Webb et al., 2023; Hodel & West, 2024), and on the use of analogy-based Chain of Thought (CoT) to improve reasoning in LLMs (Yasunaga et al., 2024; Qin et al., 2025).

In general, analogy involves drawing a parallel between two situations (Gentner, 1983; Winston, 1980), from which one tentatively infers that what holds true in the first situation may also apply to the second. When the situations come from seemingly unrelated domains, the analogy can be particularly insightful. For example, a city's transportation system can be compared to the human circulatory system. Roads, highways, and transit routes distribute people and goods throughout the city, much like blood vessels transport nutrients and oxygen through the body. Although there is no consensus on how to model the phenomenon of analogies, starting from the seminal work of Hesse (1959), a significant amount of work (Prade & Richard, 2021; 2017; Miclet & Prade, 2009; Barbot et al., 2019b; Mbengue et al., 2025; Olivier et al., 2024; Lepage & Couceiro, 2024; Marquer & Couceiro, 2024) conceives them as statements of the form $\alpha : \beta :: \gamma : \delta$, which reads "$\alpha$ is to $\beta$ as $\gamma$ is to $\delta$", with $\alpha, \beta, \gamma, \delta$ denoting logical formulas, concepts, or, more generally, multi-dimensional vectors. These are generally called proportional analogies.

In this paper, we focus on proportional analogies between vectors of binary (or Boolean) features. As shown in (Barbot et al., 2019a; Herzig et al., 2024), these vectors can be viewed as complete terms in the sense of propositional logic—that is, conjunctions of literals in which each atomic proposition appears exactly once, either as a positive or a negative literal. Following Prade & Richard (2017;

2018), we consider a difference-based interpretation of the notion of proportional analogy. Roughly speaking, according to the difference-based interpretation, a proportional analogy $\alpha : \beta :: \gamma : \delta$ means that "$\alpha$ differs from $\beta$ as $\gamma$ differs from $\delta$", and "$\beta$ differs from $\alpha$ as $\delta$ differs from $\gamma$".

The aim of this paper is to investigate the learnability of this theoretically well-founded and widely studied notion of proportional analogy, from both a theoretical and an experimental point of view. We consider both proportional analogy verification and generation. Verification involves checking whether a quadruplet $\alpha : \beta :: \gamma : \delta$ forms a valid proportional analogy, while generation involves producing a $\delta$ such that, given $\alpha$, $\beta$, and $\gamma$, the quadruplet $\alpha : \beta :: \gamma : \delta$ constitutes a valid proportional analogy.

On the theoretical side, we show that in the Boolean setting—that is, when the elements of a proportional analogy can be represented as complete terms of propositional logic corresponding to Boolean vectors—both proportional analogy verification and generation are efficiently PAC learnable.

On the experimental side, we focus on what can be described as a micro-domain of image transformations (Mitchell, 1993; Hofstadter & Mitchell, 1994), where each image is represented as a grid of cells containing shapes of various types and colors. Within this framework, we examine proportional analogies between such images. We construct both propositional logic and corresponding vector representations of proportional analogy instances. Using these vectorial forms, we automatically generate a dataset comprising valid and invalid instances of proportional analogies. This dataset is then used to train simple feedforward artificial neural networks (ANN) to perform proportional analogy verification and generation. Our results show that these ANN-based models learn efficiently, and quickly achieve a high level of accuracy.

We further compare the performance of our lightweight models, in both verification and generation, to that of state-of-the-art (SoTA) Large Language Models (LLMs) in both zero-shot and fine-tuned settings. Moreover, we evaluate their generalization capabilities relative to those of LLMs, namely their capacity to correctly verify proportional analogies in a more general setting (e.g., where images vary in both shape types and colors present in the grid cells) after being trained on a more restricted setting in which some aspects of the images played no role (e.g., colors remain fixed across the images). Our experimental results show that lightweight ANN-based models (i) are not worse than the more resource-intensive LLMs in learning proportional analogy verification and generalizing it, and (ii) outperform LLMs in learning proportional analogy generation.

### MAIN CONTRIBUTIONS AND ROADMAP

Overall, the contributions of this paper can be summarized in three points:

- Building on the performance guarantees of Couceiro et al. (2017; 2018), we show that proportional analogies are efficiently PAC-learnable in the Boolean setting.

- We generate a dataset and use it to evaluate the analogy verification and generation capabilities of simple and lightweight ANN-based models in comparison with LLMs, demonstrating the superiority of our models on these specific tasks.

- We assess the generalizability of our lightweight ANN-based models through a series of experiments on the generated dataset. Specifically, we train them on certain configurations and test it on previously unseen ones (e.g., with different colors and shapes), showing that our architectures remain competitive with LLMs both in zero-shot and fine-tuned settings.

The paper is structured as follows. Section 2 outlines the theoretical foundations of the difference-based interpretation of proportional analogy. Section 3 provides result about efficient PAC learnability of proportional analogy verification and generation. Section 4 introduces the perceptual task of identifying proportional analogies between images. In Section 5, we describe the neural network architectures for verification and generation as well as the training methodology. Section 6 presents the evaluation of our ANN-based models and their comparison with LLMs. After having discussed related work on analogical reasoning in LLMs in Section 7, we conclude.

## 2 BACKGROUND ON PROPORTIONAL ANALOGY

As pointed out in the introduction, we consider proportional analogies between complete terms. A complete term can be represented as a subset $\alpha$ of atomic propositions from a set of atomic propositions $\mathbb{P}$. The atomic propositions in $\alpha$ are the true ones (the positive literals), those in $\mathbb{P} \setminus \alpha$

are the false ones (the negative literals). For example, suppose $\mathbb{P} = \{p, q, r\}$. Then, the subset $\{p, q\}$ corresponds to the complete term $p \wedge q \wedge \neg r$.

Given four terms $\alpha, \beta, \gamma, \delta \subseteq \mathbb{P}$, we say that "$\alpha$ is to $\beta$ as $\gamma$ is to $\delta$", denoted $\alpha : \beta :: \gamma : \delta$, if

$$\alpha \setminus \beta = \gamma \setminus \delta \text{ and } \beta \setminus \alpha = \delta \setminus \gamma. \tag{1}$$

As demonstrated by Prade & Richard (2018), this definition of proportional analogy satisfies the following three properties, which are commonly considered rationality postulates for proportional analogy: *reflexivity* ($\alpha : \beta :: \alpha : \beta$), *symmetry* (if $\alpha : \beta :: \gamma : \delta$ then $\gamma : \delta :: \alpha : \beta$), and *central permutation* (if $\alpha : \beta :: \gamma : \delta$ then $\alpha : \gamma :: \beta : \delta$). It moreover satisfies the properties of *unicity* (if $\alpha : \beta :: \gamma : \delta$ and $\alpha : \beta :: \gamma : \epsilon$ then $\delta = \epsilon$), and *transitivity* (if $\alpha : \beta :: \gamma : \delta$ and $\gamma : \delta :: \epsilon : \zeta$ then $\alpha : \beta :: \epsilon : \zeta$). Moreover, as demonstrated by Herzig et al. (2024), this difference-based definition of proportional analogy between complete terms is equivalent to a formulation based on the concepts of "fore" and "back" transformations. In particular, $\alpha : \beta :: \gamma : \delta$ holds if

$$\exists \mathtt{f}_1, \mathtt{t}_1, \mathtt{f}_2, \mathtt{t}_2 \subseteq \mathbb{P} \text{ such that } \alpha \xrightarrow{\mathtt{f}_1, \mathtt{t}_1} \beta, \gamma \xrightarrow{\mathtt{f}_1, \mathtt{t}_1} \delta, \beta \xrightarrow{\mathtt{f}_2, \mathtt{t}_2} \alpha \text{ and } \delta \xrightarrow{\mathtt{f}_2, \mathtt{t}_2} \gamma,$$

where, for every $\alpha, \beta, \mathtt{f}, \mathtt{t} \subseteq \mathbb{P}$, $\alpha \xrightarrow{\mathtt{f}, \mathtt{t}} \beta$ if and only if $\beta = (\alpha \setminus \mathtt{f}) \cup \mathtt{t}$.

For example, the proportional analogy $\{p, q\} : \{p, r\} :: \{q\} : \{r\}$ holds since $\{p, q\} \xrightarrow{\{q\}, \{r\}} \{p, r\}$, $\{q\} \xrightarrow{\{q\}, \{r\}} \{r\}$, $\{p, r\} \xrightarrow{\{r\}, \{q\}} \{p, q\}$ and $\{r\} \xrightarrow{\{r\}, \{q\}} \{q\}$. The alternative way to verify the validity of the previous proportional analogy is to check that:

$$\{p, q\} \setminus \{p, r\} = \{q\} = \{q\} \setminus \{r\} \text{ and } \{p, r\} \setminus \{p, q\} = \{r\} = \{r\} \setminus \{q\}.$$

## 3 LEARNABILITY

Two problems for proportional analogy are definable: verification and generation. On the one hand, *verification* is the decision problem of checking whether, given four terms $\alpha, \beta, \gamma, \delta \subseteq \mathbb{P}$, the proportional analogy $\alpha : \beta :: \gamma : \delta$ holds. On the other hand, *generation* is the function/search problem of finding a term $\delta \subseteq \mathbb{P}$ such that, given three terms $\alpha, \beta, \gamma \subseteq \mathbb{P}$, the proportional analogy $\alpha : \beta :: \gamma : \delta$ holds. We enumerate the atomic propositions in $\mathbb{P}$ by means of a bijection $e : \{1, \ldots, |\mathbb{P}|\} \longrightarrow \mathbb{P}$. We are going to use this enumeration to provide a vector representation of terms.

It is straightforward to see that the verification problem is equivalent to the problem of computing the function $f_{verif} : (\{0, 1\}^4)^{|\mathbb{P}|} \longrightarrow \{0, 1\}$, such that for all $\mathbf{x}_1, \ldots, \mathbf{x}_{|\mathbb{P}|} \in \{0, 1\}^4$,

$$f_{verif}(\mathbf{x}_1, \ldots, \mathbf{x}_{|\mathbb{P}|}) = 1 \text{ iff } \forall i \in \{1, \ldots, |\mathbb{P}|\}, f_{loc}(\mathbf{x}_i) = 1,$$

where $f_{loc}(\mathbf{x}_i) = 1$ iff $\mathbf{x}_i(1) \cdot (1 - \mathbf{x}_i(2)) = \mathbf{x}_i(3) \cdot (1 - \mathbf{x}_i(4))$ and $\mathbf{x}_i(2) \cdot (1 - \mathbf{x}_i(1)) = \mathbf{x}_i(4) \cdot (1 - \mathbf{x}_i(3))$, with $\mathbf{x}_i(j)$ being the $j$-th bit of the 4-bit vector $\mathbf{x}_i$.

Notice that the function $f_{verif}$ is not linearly separable. To see this, consider the case $|\mathbb{P}| = 1$ and observe that $f_{verif}((0), (0), (0), (0)) = f_{verif}((1), (1), (1), (1)) = 1$, while $f_{verif}((0), (1), (1), (0)) = 0$. Thus, the function $f_{verif}$ is not learnable by a single-layer perceptron.

The connection between the verification problem for analogical proportion and the function $f_{verif}$ is highlighted by the following proposition.

**Proposition 1.** *Let $\alpha_1, \alpha_2, \alpha_3, \alpha_4 \subseteq \mathbb{P}$. Then,*

$$\alpha_1 : \alpha_2 :: \alpha_3 : \alpha_4 \text{ holds iff } f_{verif}(\mathbf{x}_1^{(\alpha_1, \alpha_2, \alpha_3, \alpha_4)}, \ldots, \mathbf{x}_{|\mathbb{P}|}^{(\alpha_1, \alpha_2, \alpha_3, \alpha_4)}) = 1,$$

*where, for all $i \in \{1, \ldots, |\mathbb{P}|\}$ and all $j \in \{1, \ldots, 4\}$, we have $\mathbf{x}_i^{(\alpha_1, \alpha_2, \alpha_3, \alpha_4)}(j) = 1$ iff $e(i) \in \alpha_j$.*

*Proof.* See Appendix A.1. $\qquad\square$

Consequently, the generation problem is equivalent to the problem of computing the following function $f_{gen} : (\{0, 1\}^3)^{|\mathbb{P}|} \longrightarrow 2^{(\{0,1\}^{|\mathbb{P}|})}$, such that for all $\mathbf{x}_1, \ldots, \mathbf{x}_{|\mathbb{P}|} \in \{0, 1\}^3$ and $\mathbf{y} \in \{0, 1\}^{|\mathbb{P}|}$

$$\mathbf{y} \in f_{gen}(\mathbf{x}_1, \ldots, \mathbf{x}_{|\mathbb{P}|}) \text{ iff } f_{verif}((\mathbf{x}_1, \mathbf{y}(1)), \ldots, (\mathbf{x}_{|\mathbb{P}|}, \mathbf{y}(|\mathbb{P}|))) = 1,$$

where for every $i \in \{1, \ldots, |\mathbb{P}|\}$, $(\mathbf{x}_i, \mathbf{y}(i))$ is the extension of the Boolean vector $\mathbf{x}_i$ with the scalar Boolean value $\mathbf{y}(i)$.

It is straightforward to see that the set $f_{gen}(\mathbf{x}_1, \ldots, \mathbf{x}_{|\mathbb{P}|})$ is a singleton. This is related to the unicity property of proportional analogy, we mentioned in Section 2. Thus, the codomain of the function $f_{gen}$ can be seen as $\{0,1\}^{|\mathbb{P}|}$ instead of $2^{\left(\{0,1\}^{|\mathbb{P}|}\right)}$.

The connection between the generation problem and the function $f_{gen}$ is highlighted by the following proposition. We do not give the proof since it is analogous to the proof of Proposition 1.

**Proposition 2.** *Let $\alpha_1, \alpha_2, \alpha_3, \alpha_4 \subseteq \mathbb{P}$. Then,*

$$\alpha_1 : \alpha_2 :: \alpha_3 : \alpha_4 \text{ holds iff } \mathbf{y}^{\alpha_4} \in f_{gen}(\mathbf{x}_1^{(\alpha_1, \alpha_2, \alpha_3)}, \ldots, \mathbf{x}_{|\mathbb{P}|}^{(\alpha_1, \alpha_2, \alpha_3)}),$$

*where, for all $i \in \{1, \ldots, |\mathbb{P}|\}$ and for all $j \in \{1, 2, 3\}$,*

$$\mathbf{x}_i^{(\alpha_1, \alpha_2, \alpha_3)}(j) = 1 \text{ iff } e(i) \in \alpha_j, \text{ and } \mathbf{y}^{\alpha_4}(i) = 1 \text{ iff } e(i) \in \alpha_4.$$

We conclude this section with a proof of the efficient PAC learnability of the functions $f_{verif}$ and $f_{gen}$. We remind the notion of efficient PAC learnability (Mohri et al., 2018). A function is *PAC learnable* if there is an algorithm that, given random labeled examples from any input distribution, can, with probability at least $1 - \delta$, produce a hypothesis whose error is at most $\varepsilon$ using only a polynomial number of samples in $1/\varepsilon$ and $1/\delta$. It is *efficiently PAC learnable* if the algorithm also runs in polynomial time with respect to the input size, $1/\varepsilon$, and $1/\delta$. In short, PAC learnability guarantees data efficiency, while efficient PAC learnability guarantees both data and time efficiency.

**Theorem 1.** *The functions $f_{verif}$ and $f_{gen}$ are both efficiently PAC learnable.*

*Proof.* See Appendix A.2. □

Theorem 1, together with Propositions 1 and 2, shows that both proportional analogy verification and generation are efficiently PAC learnable.

## 4 PERCEPTIVE TASK

In this section, we introduce the perceptual scenario that will be used to evaluate the ability of a trained artificial neural network to verify and generate proportional analogies. In this scenario, a proportional analogy is defined between images of a grid whose cells may contain shapes of various colours. In particular, we consider a $n \times n$ grid, $k$ possible shape types, and $m$ possible colours. Our logical encoding of the scenario requires the following set of atomic propositions:

$$\mathbb{P} = \big\{ \mathtt{t}_{x,y,z} : x, y \in D \text{ and } z \in T \big\} \cup \big\{ \mathtt{c}_{x,y,z} : x, y \in D \text{ and } z \in C \big\},$$

where $D = \{1, \ldots, n\}$, $T = \{t_1, \ldots, t_k\}$, and $C = \{c_1, \ldots, c_m\}$ are, respectively, the set of grid dimensions, the set of shape types, and the set of colours. The atomic proposition $\mathtt{t}_{x,y,z}$ stands for "there is a shape of type $z$ in the cell $(x, y)$ of the grid", while the atomic proposition $\mathtt{c}_{x,y,z}$ stands for "there is a shape of colour $z$ in the cell $(x, y)$ of the grid". It is easy to verify that $|\mathbb{P}| = |D| \times |D| \times |T| \times |C|$. For example, in the scenario of a $2 \times 2$ grid with 2 possible shape types and 2 possible colours, we have $|\mathbb{P}| = 16$. Thus, to represent the image of the grid, we need a vector with 16 binary dimensions. In the scenario of a $2 \times 2$ grid with 2 possible shape types and 3 possible colours, $|\mathbb{P}| = 24$. Hence, in this case, to represent the image of the grid, we need a vector with 24 binary dimensions.

Figure 1 provides an instance of proportional analogy in the case of a $2 \times 2$ grid, with a set of two shape types $T = \{t, c\}$, and set of three colours $C = \{r, g, b\}$, where $t, c, r, g$ and $b$ stand, respectively, for 'triangle', 'circle', 'red', 'green' and 'blue'. The proportional analogy shown in the figure can be formally represented as follows:

$$\{\mathtt{t}_{2,1,t}, \mathtt{c}_{2,1,g}\} : \{\mathtt{t}_{2,1,t}, \mathtt{c}_{2,1,r}, \mathtt{t}_{1,1,c}, \mathtt{c}_{1,1,r}, \mathtt{c}_{1,2,t}, \mathtt{c}_{1,2,b}\} ::$$
$$\{\mathtt{t}_{2,1,c}, \mathtt{c}_{2,1,g}\} : \{\mathtt{t}_{2,1,c}, \mathtt{c}_{2,1,r}, \mathtt{t}_{1,1,c}, \mathtt{c}_{1,1,r}, \mathtt{c}_{1,2,t}, \mathtt{c}_{1,2,b}\} \quad (2)$$

It is routine to verify that this is a valid proportional analogy according to the difference-based interpretation formally defined in Section 2.

We assume that not all images are admissible. In particular, each cell in the grid can either be empty or contain a single shape type in a given color. Consequently, in this perceptual scenario, and in the corresponding dataset described in Section 5, the terms $\alpha, \beta, \gamma$, and $\delta$ of a proportional analogy $\alpha : \beta :: \gamma : \delta$ can only be generated from the following set, which is a strict subset of $2^{\mathbb{P}}$:

$$
\begin{aligned}
Term = \big\{ \alpha \subseteq \mathbb{P} : & (\forall x, y \in D, \forall z \in T, \text{ if } \mathtt{t}_{x,y,z} \in \alpha \text{ then } \exists z' \in C \text{ s.t. } \mathtt{c}_{x,y,z'} \in \alpha), \\
& (\forall x, y \in D, \forall z \in T, \text{ if } \mathtt{c}_{x,y,z} \in \alpha \text{ then } \exists z' \in T \text{ s.t. } \mathtt{t}_{x,y,z'} \in \alpha), \\
& (\forall x, y \in D, \forall z, z' \in T \text{ if } \mathtt{t}_{x,y,z}, \mathtt{t}_{x,y,z'} \in \alpha \text{ then } z = z'), \\
& (\forall x, y \in D, \forall z, z' \in C \text{ if } \mathtt{c}_{x,y,z}, \mathtt{c}_{x,y,z'} \in \alpha \text{ then } z = z') \big\}
\end{aligned}
$$

The cardinality of the set $Term$ can be easily computed by noting that each cell in the grid admits the following number of possible configurations:

$$\mathsf{n.cell} = (|T| \times |C|) + 1 \tag{3}$$

Indeed, a cell can either be empty or contain a shape from $T$ in a colour from $C$. Thus, the following is the number of possible images of the grid:

$$\mathsf{n.img} = \mathsf{n.cell}^{|D| \times |D|} \tag{4}$$

The number $\mathsf{n.img}$ corresponds to the cardinality of the set $Term$. Since a proportional analogy is a quadruplet of images of the grid (i.e., of elements of $Term$), the following is the number of possible proportional analogies that can be generated:

$$\mathsf{n.ap} = \mathsf{n.img}^4 \tag{5}$$

For example, in the case of a $2 \times 2$ grid, with a set of two shape types and a set of three colours, approximately $33 \times 10^{12}$ proportional analogies can be generated.

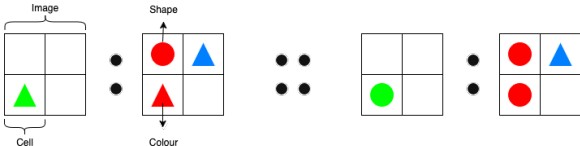

Figure 1: A valid instance of proportional analogy

## 5 NEURAL ARCHITECTURE AND DATASET

In this section, we present how the dataset was constructed, along with the neural network architectures used for proportional analogy verification and generation.

### 5.1 DATASET

To generate the data required for the experiments, we first define the set of shape types $T$ and colors $C$, following the example in Figure 1, which features $D = 2$ dimensions and therefore $2 \times 2$ grids. We then generate images by combining these shape types and colors, obtaining the complete set of images. Proportional analogies are formed as 4-tuples of images. For the experiments, we constructed two datasets of different scales. The smaller dataset uses $T = \{s\}$ and $C = \{r, g\}$, while the larger one uses $T = \{s, t\}$ and $C = \{r, g, b\}$. The counts of cells, images, and proportional analogies for each configuration are summarized in Table 5 given in Appendix A.3. From Equations 3, 4, and 5, it is clear that the dataset size grows exponentially with the number of shapes and colors. To keep the dataset manageable, we restrict ourselves to using up to 2 shapes (square, triangle) and 3 colors (red, green, blue), which represents a reasonable compromise. The generated datasets, along with the code used to create them, are publicly available in our repository.[1] For the configuration

---

[1]The repository will be updated upon acceptance.

with $T = \{s\}$ and $C = \{r, g\}$, all 81 images are used to generate every possible proportional analogy. Since the majority of these are invalid proportional analogies, we balance the dataset by including all valid proportional analogies and randomly sampling an equal number of negative ones. This results in the balanced *ds12* dataset containing 183,290 instances. For the configuration with $T = \{s, t\}$ and $C = \{r, g, b\}$, generating proportional analogies from all 2,401 images would be computationally prohibitive. Instead, we randomly sample 81 images—the same number used in *ds12*— to generate proportional analogies, then we apply the same balancing procedure, yielding the *ds23* dataset with 29,210 instances. The smaller size of *ds23*, despite using the same number of images to produce proportional analogy instances, is due to a lower proportion of valid instances as the sets of shape types and colours expand. In all experimental settings, we follow the widely accepted train–validation–test split of 70%–10%–20%. The number of instances after balancing for each configuration is reported in Table 6 given in Appendix A.3.

## 5.2 MODEL FOR VERIFICATION

As part of this study, which aims to provide insights into how accurately and how quickly a lightweight artificial neural network can learn proportional analogies, we strive to keep the experimental setup as intuitive as possible. The objective is to develop a model capable of automatically recognizing valid and invalid instances of proportional analogies, based on the difference-based interpretation. To this end, we design, train, and evaluate a binary classifier that takes the vector representation of a proportional analogy instance as input and outputs whether the instance is valid or invalid.

We employ a simple feed-forward architecture consisting of a few linear layers, with dropout and activation functions placed between them. We experimented with different numbers of linear layers, aiming to keep the architecture as small as possible without sacrificing performance. The overall design of the model is shown in Figure 2: the model for proportional analogy verification is depicted on the left, while the one for proportional analogy generation, discussed in Section 5.3, is on the right. In line with what we explained in Section 3, we produce one-hot representations in

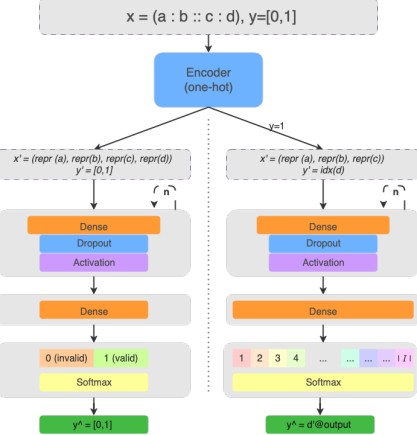

Figure 2: Pipeline architecture. From left to right: model for verification, model for generation

the form of a fixed size quadruplets of Boolean vectors. For example, the following quadruplet of Boolean vectors correponds to the valid proportional analogy of Figure 1:

$$[0, 0, 0, 0, 0, 0, 0, 0, 0, 0, 1, 0, 0, 1, 0, 0, 0, 0, 0, 0] : [0, 1, 1, 0, 0, 1, 0, 0, 0, 1, 1, 0, 1, 0, 0, 0, 0, 0, 0, 0] ::$$
$$[0, 0, 0, 0, 0, 0, 0, 0, 0, 0, 0, 1, 0, 1, 0, 0, 0, 0, 0, 0] : [0, 1, 1, 0, 0, 1, 0, 0, 0, 1, 0, 1, 1, 0, 0, 0, 0, 0, 0, 0] \quad (6)$$

Booleans are transformed into real values and the vectors are fed as input to the ANN. Using the encoded vectors from the train–dev–test split, we train a binary classification model with valid proportional analogies represented by 1, or 0 otherwise. We experimented with various hyperparameters to optimize the training process, including number of intermediate blocks, dropout rate, feature reduction function for linear transformations, and activation function, all defined within a parameter grid. Our experiments indicate that optimal training is achieved with 2 intermediate blocks, 0.3 dropout rate, geometric progression function for feature reduction, and Rectified Linear Unit (ReLU) as activation function. Additionally, a learning rate of $10^{-3}$ and a batch size of 64 were

used during training. Overall, the models converged rapidly to near-perfect training accuracy within fewer than 10 epochs, using a learning rate of $10^{-3}$ for both the models trained on $ds12$ and $ds23$. The corresponding training accuracy and loss curves are presented in Figure 3 in the Appendix.

## 5.3 MODEL FOR GENERATION

Since we frame our generative task as a proportional analogy completion task, the model is designed to predict the vector representation of a fourth image given the vector representations of a triplet of three images. We employ a feed-forward neural network architecture for sequence generation, as illustrated in Figure 2. The network consists of multiple fully connected (dense) layers with ReLU activation functions, with dropout regularization applied between hidden layers to mitigate overfitting. The final layer uses a Softmax activation to produce a probability distribution over the target vocabulary. During training, the model is optimized to maximize the likelihood of the correct next token given the input sequence, using cross-entropy loss. In our setup, the vocabulary size corresponds to the discrete set of possible image tokens generated by our system's image tokenization scheme, denoted as n.img in Table 5 given in Appendix A.3. We adapt datasets $ds12$ and $ds23$, originally constructed for the verification task, introducing modifications to obtain $ds12\_g$ and $ds23\_g$ for the generation task. Each valid proportional analogy $\alpha : \beta :: \gamma : \delta$ is tokenized into individual elements $\{\alpha, \beta, \gamma, \delta\}$, with the first three elements $(\alpha, \beta, \gamma)$ forming the input sequence and $\delta$ serving as the target output. For each dataset, the vocabulary is constructed using our original image generator. This strategy reduces unwanted bias and promotes generalizability, as the vocabulary remains fixed and is not limited to the subset of images appearing in the proportional analogies. The train, validation, and test splits remain strictly disjoint. Dataset statistics are presented in Table 7. We train two separate models for this task, corresponding to $ds12\_g$ and $ds23\_g$ datasets. Each model receives an input sequence of three tokens, with each token representing an image encoded as a one-hot vector. Model validation is performed on the development set after every epoch, and optimization is performed using cross-entropy loss. Empirically, we find that architectures with two dense (fully connected) layers yield the best results. The optimal hyperparameters, determined on the validation set, are a learning rate of $5 \times 10^{-3}$ over 30 epochs. Training progresses rapidly, with models achieving high accuracy after only a few epochs and converging to near-perfect scores by the end of training.

## 6 EVALUATION

We compare our approach with LLMs by evaluating performance on the two tasks: proportional analogy verification and generation.

**Zero-shot classification** As *Llama-3.1-8B-Instruct* operates based on instruction prompts, we begin by providing it with the following task-specific instruction:

> *"Given the descriptions of four images in the form of binary values representing boolean features, answer only 'yes' or 'no' to indicate whether they form a valid analogy. Do not provide any further explanation."*

Once the instruction is set, the representations of our analogical instances are passed to the model. A response of "yes" is interpreted as identifying a *valid* proportional analogy, while "no" indicates an *invalid* one. When compared to our model, *Llama-3.1-8B-Instruct* underperforms significantly. Despite the binary nature of the classification task, its highest accuracy reaches only 55.73%, whereas our model achieves near-perfect results. Furthermore, across all other evaluation metrics—precision, recall, and F1 score—our feed-forward neural network consistently and substantially outperforms the large language model, regardless of the experimental setting. As we move from the $ds12$ to the $ds23$ dataset, the number of features in the input vectors increases, and both models show improved performance. This trend is promising for future work and warrants further investigation into how input complexity affects model performance in analogical reasoning tasks.

**Fine-tuned models for verification** We selected several open-source and state-of-the-art models from the MTEB — Multilingual Text Embedding Benchmark (Muennighoff et al., 2022) leaderboard for classification tasks and fine-tuned them for the sequence classification setting. The models

| Dataset | Model | Precision | Recall | F1 score | Accuracy |
|---------|-------|-----------|--------|----------|----------|
| *ds12* | *Llama-3.1-8B-Instruct* | 53.11/79.64 | 95.97/15.69 | 68.38/26.21 | 55.73 |
| *ds12* | Ours | 100.0/99.88 | 99.88/100.0 | 99.94/99.94 | 99.94 |
| *ds23* | *Llama-3.1-8B-Instruct* | 62.21/73.53 | 81.84/50.38 | 70.69/59.79 | 66.09 |
| *ds23* | Ours | 100.0/99.97 | 99.97/100.0 | 99.98/99.98 | 99.98 |

Table 1: Evaluation scores of zero-shot *Llama-3.1-8B-Instruct* vs our model

were chosen not only for their overall performance but also for their comparable number of parameters, ensuring a fairer comparison. To improve computational efficiency, we applied Quantized Low-Rank Adaptation (QLoRA) (Dettmers et al., 2023) with 4-bit NormalFloat (NF4) quantization during fine-tuning. Table 2 presents the results obtained with the fine-tuned models.

| Dataset | Model | Precision | Recall | F1 score | Accuracy |
|---------|-------|-----------|--------|----------|----------|
| *ds12* | *Llama-3.1-8B* | 98.18/96.40 | 96.31/98.22 | 97.24/97.30 | 97.27 |
| *ds12* | *Mistral-7B-v0.3* | 99.96/99.80 | 99.80/99.96 | 99.88/99.88 | 99.88 |
| *ds12* | *Qwen3-8B* | 99.70/99.32 | 99.31/99.70 | 99.50/99.51 | 99.51 |
| *ds12* | Ours | 100.0/99.88 | 99.88/100.0 | 99.94/99.94 | 99.94 |
| *ds23* | *Llama-3.1-8B* | 99.06/98.73 | 98.72/99.07 | 98.89/98.90 | 98.90 |
| *ds23* | *Mistral-7B-v0.3* | 98.57/99.89 | 99.89/98.55 | 99.23/99.22 | 99.23 |
| *ds23* | *Qwen3-8B* | 99.41/99.59 | 99.59/99.41 | 99.50/99.50 | 99.50 |
| *ds23* | Ours | 100.0/99.97 | 99.97/100.0 | 99.98/99.98 | 99.98 |

Table 2: Evaluation scores on fine-tuned LLMs for analogy verification vs our model

The results indicate that pretrained LLMs, once fine-tuned, can tackle the proportional analogy verification task with strong performance, though slightly underperform compared to our neural network models. When taking the model sizes into consideration, our models of less than 1000 parameters punch well above its weight class.

**Fine-tuned models for generation** Using the same large language models selected for the verification task, we fine-tune them to predict the missing element $\delta$ that completes a valid analogy $\alpha : \beta :: \gamma : \delta$, given $(\alpha, \beta, \gamma)$ as input. This setup mirrors the training objective of our feed-forward model for generation. All models are trained, validated, and evaluated on identical train–dev–test splits. Table 3 reports their performance.

| Dataset | Model | Precision | Recall | F1 score | Accuracy |
|---------|-------|-----------|--------|----------|----------|
| *ds12_g* | *Llama-3.1-8B* | 99.8896 | 99.8997 | 99.8933 | 99.8965 |
| *ds12_g* | *Mistral-7B-v0.3* | 98.8851 | 99.0667 | 98.8142 | 99.1505 |
| *ds12_g* | *Qwen3-8B* | 98.7442 | 98.4103 | 98.5464 | 99.1068 |
| *ds12_g* | Ours | 99.9904 | 99.9179 | 99.9537 | 99.9673 |
| *ds23_g* | *Llama-3.1-8B* | 75.8494 | 67.5725 | 67.2116 | 69.8936 |
| *ds23_g* | *Mistral-7B-v0.3* | 97.7164 | 97.5606 | 97.5969 | 97.6313 |
| *ds23_g* | *Qwen3-8B* | 69.3315 | 59.3813 | 60.3614 | 61.3457 |
| *ds23_g* | Ours | 99.8792 | 99.8578 | 99.8636 | 99.8627 |

Table 3: Evaluation scores on fine-tuned LLMs for generation vs our model

Table 3 demonstrates that large language models (LLMs) achieve strong performance in predicting the element $\delta$ on the *ds12_g* dataset. However, as the number of properties increases, leading to an exponential growth in the number of classes, as seen in *ds23_g*, most LLMs experience a significant decline in performance, with the exception of *Mistral-7B-v0.3*. In contrast, our model continues to perform competitively under these more challenging conditions.

**Generalisability** To rigorously evaluate the generalisability of the models in scenarios involving instances with previously unobserved properties, we construct novel datasets. In these scenarios, all

proportional analogies lacking either a specified shape or colour are excluded. Specifically, we generate two datasets, denoted *ds23_square* and *ds23_red*. These datasets are employed both for training our model and for fine-tuning several large language models, which are subsequently evaluated on the complete test set of $ds23$. Model training and fine-tuning are conducted using vector representations of proportional analogies as the primary input. The resulting classification performance is reported in Table 4. The results showcased the strong performance of both LLMs and our models in terms of generalisability. In both scenarios, the models generalise well from a smaller to a greater domain without catastrophic degradation of performance, demonstrated by a marginal decrease in the scores.

| Train set | Model | Precision | Recall | F1 score | Accuracy |
|---|---|---|---|---|---|
| *ds23_square* | *Llama-3.1-8B* | 98.92/100.0 | 100.0/98.90 | 99.46/99.45 | 99.45 |
| *ds23_square* | *Mistral-7B-v0.3* | 99.63/99.83 | 99.83/99.62 | 99.73/99.73 | 99.73 |
| *ds23_square* | *Qwen3-8B* | 99.78/92.52 | 91.98/99.79 | 95.72/96.01 | 95.87 |
| *ds23_square* | Ours | 99.75/95.69 | 95.53/99.76 | 97.59/97.68 | 97.64 |
| *ds23_red* | *Llama-3.1-8B* | 98.82/100.0 | 100.0/98.80 | 99.41/99.39 | 99.40 |
| *ds23_red* | *Mistral-7B-v0.3* | 98.38/99.83 | 99.83/98.35 | 99.10/99.08 | 99.09 |
| *ds23_red* | *Qwen3-8B* | 98.45/99.62 | 99.62/98.42 | 99.03/99.01 | 99.02 |
| *ds23_red* | Ours | 99.96/95.07 | 94.84/99.97 | 97.34/97.46 | 97.40 |

Table 4: Results on analogy verification and their generalisability. Test results are on *ds23* test set.

## 7 RELATED WORK ON ANALOGICAL REASONING IN LLMS

Several studies have investigated the analogical reasoning capabilities of large language models (LLMs). For example, Webb et al. (2023) evaluate GPT-3 (Brown et al., 2020) on four types of analogical reasoning tasks: Raven's Progressive Matrices (Raven, 1965), letter-string analogies in the CopyCat micro-domain (Hofstadter, 1984; Hofstadter & Mitchell, 1994; Mitchell, 1993), proportional analogies on verbal expressions, and story analogies (Gentner et al., 1993). Raven's Matrices are encoded using different formalisms, which, according to the authors, ensures the zero-shot nature of the task since the model has never encountered the input data. Their findings show that GPT-3 achieves results comparable to, and in some cases better than, human performance. In a different line of research, Yasunaga et al. (2024) introduce analogical prompting within a Chain-of-Thought (CoT) framework. They test various LLMs—including GPT-3.5-turbo, text-davinci-003 (Ouyang et al., 2022), and PaLM 2 (Anil et al., 2023)—on coding, mathematical, and reasoning tasks, where the models are explicitly prompted to recall analogous problems before solving the target problem. Their approach is compared against standard CoT methods. Results show notable improvements when LLMs are guided to generate their own analogous exemplars prior to problem solving.

## 8 CONCLUSION

To summarize, we have presented theoretical results on the efficient PAC learnability of proportional analogy verification and generation in the Boolean setting. Additionally, we trained simple artificial neural networks (ANNs) to both verify and generate proportional analogies. Our findings demonstrate that these ANNs can learn these tasks quickly and with high accuracy. We also compared the performance of our lightweight models with that of state-of-the-art large language models (LLMs), both in a zero-shot setting and after fine-tuning, with respect to: (i) proportional analogy verification, (ii) proportional analogy generation, and (iii) the generalizability of proportional analogy verification. Overall, our experiments indicate that proportional analogies can be learned efficiently by lightweight ANN-based models, and that the knowledge acquired by these models can also be generalized effectively, with no clear advantage offered by energy-intensive LLMs in this context. In line with recent work exploring the connection between PAC learnability and deep learning models (Ghojogh & Ghodsi, 2024), in future work we plan to study in depth the relationship between our efficient PAC learnability results and the neural network models we implemented. In particular, the research question we aim to address in future work is: What is the minimal number of parameters that feed-forward neural network architectures, like those presented in Section 5, must have to achieve high accuracy in verifying and generating proportional analogies in polynomial time?

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

# A    APPENDIX

## A.1    PROOF OF PROPOSITION 1

**Proposition 1.** *Let* $\alpha_1, \alpha_2, \alpha_3, \alpha_4 \subseteq \mathbb{P}$. *Then,*

$$\alpha_1 : \alpha_2 :: \alpha_3 : \alpha_4 \text{ holds iff } f_{verif}(\mathbf{x}_1^{(\alpha_1,\alpha_2,\alpha_3,\alpha_4)}, \dots, \mathbf{x}_{|\mathbb{P}|}^{(\alpha_1,\alpha_2,\alpha_3,\alpha_4)}) = 1,$$

*where, for all* $i \in \{1, \dots, |\mathbb{P}|\}$ *and all* $j \in \{1, \dots, 4\}$, *we have* $\mathbf{x}_i^{(\alpha_1,\alpha_2,\alpha_3,\alpha_4)}(j) = 1$ *iff* $e(i) \in \alpha_j$.

*Proof.* Note that $\mathbf{x}_i^{(\alpha_1,\alpha_2,\alpha_3,\alpha_4)}$ is the 4-dimensional Boolean vector corresponding to the proposition $e(i)$ in $\mathbb{P}$: a position $j$ of the vector with $j \in \{1, \dots, 4\}$ has value 1 iff the proposition $e(i)$ belongs to the term $\alpha_j$ in the proportional analogy.

The fact that $\alpha_1 : \alpha_2 :: \alpha_3 : \alpha_4$ holds means that $\alpha_1 \setminus \alpha_2 = \alpha_3 \setminus \alpha_4$ and $\alpha_2 \setminus \alpha_1 = \alpha_4 \setminus \alpha_3$. The latter is equivalent to say that, for every $p \in \mathbb{P}$: i) ($p \in \alpha_1$ and $p \notin \alpha_2$) iff ($p \in \alpha_3$ and $p \notin \alpha_4$), and ii) ($p \notin \alpha_1$ and $p \in \alpha_2$) iff ($p \notin \alpha_3$ and $p \in \alpha_4$). The latter is equivalent to say that, for every $p \in \mathbb{P}$:

$$\mathbf{x}_{e^{-1}(p)}^{(\alpha_1,\alpha_2,\alpha_3,\alpha_4)}(1) \cdot \left(1 - \mathbf{x}_{e^{-1}(p)}^{(\alpha_1,\alpha_2,\alpha_3,\alpha_4)}(2)\right) =$$
$$\mathbf{x}_{e^{-1}(p)}^{(\alpha_1,\alpha_2,\alpha_3,\alpha_4)}(3) \cdot \left(1 - \mathbf{x}_{e^{-1}(p)}^{(\alpha_1,\alpha_2,\alpha_3,\alpha_4)}(4)\right),$$

and

$$\mathbf{x}_{e^{-1}(p)}^{(\alpha_1,\alpha_2,\alpha_3,\alpha_4)}(2) \cdot \left(1 - \mathbf{x}_{e^{-1}(p)}^{(\alpha_1,\alpha_2,\alpha_3,\alpha_4)}(1)\right) =$$
$$\mathbf{x}_{e^{-1}(p)}^{(\alpha_1,\alpha_2,\alpha_3,\alpha_4)}(4) \cdot \left(1 - \mathbf{x}_{e^{-1}(p)}^{(\alpha_1,\alpha_2,\alpha_3,\alpha_4)}(3)\right),$$

where $e^{-1}$ is the inverse of the function $e$. The latter is equivalent to $f_{verif}(\mathbf{x}_1^{(\alpha_1,\alpha_2,\alpha_3,\alpha_4)}, \dots, \mathbf{x}_{|\mathbb{P}|}^{(\alpha_1,\alpha_2,\alpha_3,\alpha_4)}) = 1$. □

## A.2    PROOF OF THEOREM 1

**Theorem 1.** *The functions* $f_{verif}$ *and* $f_{gen}$ *are both efficiently PAC learnable.*

*Proof.* The target function $f_{verif}$ to be learned takes $4|\mathbb{P}|$ input bits and is defined as the conjunction of $|\mathbb{P}|$ identical copies of a 4-bit Boolean function $f_{loc}$. The hypotheses to be considered are therefore all Boolean functions with domain $\{0,1\}^4$ and codomain $\{0,1\}$, of which $f_{loc}$ is the true instance. There are exactly $2^{16} = 65{,}536$ such candidate functions.

Since the domain of $f_{loc}$ has size 16, at most 16 distinct labeled examples of the form $(Y, \tau)$ with $Y \in \{0,1\}^4$ would be sufficient to uniquely identify $f_{loc}$ with probability 1 and zero error. However, the available training examples are of the form $(X, \tau)$ with $X \in (\{0,1\}^4)^{|\mathbb{P}|}$, so each example provides information about multiple blocks simultaneously. Two different examples may cover the same 4-bit inputs, so in practice more than 16 full examples are needed to observe all 16 inputs with high probability. By a straightforward probabilistic argument,[2] the number of examples needed to guarantee that any remaining consistent hypothesis has error at most $\epsilon$ with probability at least $1 - \delta$ is polynomial in $|\mathbb{P}|$, $1/\epsilon$, and $1/\delta$. Let us denote this number by $m$.

A simple brute-force algorithm enumerates all candidate functions, applies each to every 4-bit block of an example, takes the conjunction, and discards any inconsistent hypotheses. Evaluating a candidate on one example requires $O(|\mathbb{P}|)$ operations, and with $m$ examples the total runtime is $O(m \cdot |\mathbb{P}|)$, which is polynomial in the relevant parameters.

Proving that the function $f_{gen}$ is efficiently PAC learnable can be proved analogously. Indeed, the target function $f_{gen}$ to be learned takes $3|\mathbb{P}|$ input bits and is defined as the conjunction of $|\mathbb{P}|$ identical copies of a 3-bit Boolean function. □

---

[2]This is analogous to the coupon-collector problem: each of the 16 possible 4-bit inputs can be seen as a distinct "coupon," and each training example provides $|\mathbb{P}|$ random coupons (the blocks it contains). Standard results show that, to see all 16 coupons with high probability, a number of examples polynomial in $|\mathbb{P}|$ suffices.

## A.3 DATASET STATISTICS

| $|T|, |C|$ | 1, 2 | 2, 3 |
|---|---|---|
| n.cell | 3 | 7 |
| n.img | 81 | 2,401 |
| n.ap | $\approx 43 \times 10^6$ | $\approx 33 \times 10^{12}$ |

Table 5: Counts of cells, images, and proportional analogies as a function of the number of shapes and colors

| Dataset | $|T|, |C|$ | all | train | dev | test |
|---|---|---|---|---|---|
| *ds12* | 1, 2 | 183 290 | 128 303 | 18 329 | 36 658 |
| *ds23* | 2, 3 | 29 210 | 20 447 | 2 921 | 5 842 |

Table 6: Number of instances for the verification task across experimental settings

| Dataset | $|T|, |C|$ | all | train | dev | test |
|---|---|---|---|---|---|
| *ds12_g* | 1, 2 | 91,645 | 64,200 | 9,082 | 18,363 |
| *ds23_g* | 2, 3 | 14,605 | 10,213 | 1,479 | 2,913 |

Table 7: Number of instances for the generation task across experimental settings

## A.4 ANALOGICAL MODEL TRAINING SCORES

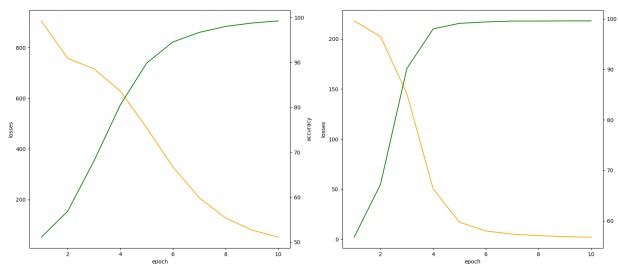

Figure 3: Training accuracy and loss for the model trained on $ds12$ (left) and $ds23$ (right) datasets, over 10 epochs each.

