# OpenReview forum: "Learning Proportional Analogies: Lightweight Neural Network vs LLM"
_ICLR.cc/2026/Conference — ICLR 2026 Conference Withdrawn Submission_

### Official Review · Reviewer_RtCQ · 2025-10-31

**Soundness:** 2
**Presentation:** 3
**Contribution:** 2
**Rating:** 4
**Confidence:** 4

**Summary:**

The paper studies proportional analogies (α:β::γ:δ), proves efficient PAC learnability for verification and generation in the Boolean setting, and evaluates tiny feed-forward ANNs against LLMs on synthetic “4-cell image” analogies (verification, generation, and generalization). The ANN matches/edges out fine-tuned 7–8B LLMs and dominates zero-shot LLMs, with near-perfect accuracy and minimal parameters.

**Strengths:**

1. Sound theory. The proof of Theorem 1 is clear and convincing.
2. Substantial evaluation data. The experiments use a large dataset to test both verification and generation.
3. Clear presentation. The paper is well organized and easy to follow.

**Weaknesses:**

1. Narrow baselines. Missing few-shot setups, closed-source models, and reasoning-focused LLMs; the comparison set is limited.
2. Ceiling effects. After fine-tuning, large models perform on par with the small network (near 100%), suggesting the task may be too easy; consider harder variants or other domains (e.g., text-only analogies).
3. Limited scaling tests. No results with bigger grids or more attributes (more shapes/colors).

**Questions:**

1. Baselines: Can you add few-shot prompts, strong closed-source models, and reasoning-centric LLMs to the comparisons?
2. Task difficulty: How does performance change on harder settings (larger grids, more shapes/colors, added noise)?
3. Domain generalization: Do the methods transfer to other analogy forms, such as pure text analogies?
4. Scaling/complexity: Have you tried systematic scaling studies (e.g., 3×3 grids, more attributes) to find the breakpoints?

---

### Official Review · Reviewer_58dL · 2025-10-31

**Soundness:** 3
**Presentation:** 3
**Contribution:** 2
**Rating:** 2
**Confidence:** 4

**Summary:**

This paper investigates the learnability of proportional analogies a:b::c:d where each item (a, b, c, d) is a set of features represented in 1-hot vector format.  The paper considers two tasks:
(1) given a, b, c, d, classify the analogy as "valid" (as defined in the paper) or “invalid”
(2) given a, b, c, generate a d that results in valid analogy

The authors did the following:
1. Gave a mathematical proof that these tasks are PAC-learnable

2. Generated dataset of valid and invalid proportional analogies

3. Showed that relatively small feedforward DNNs can learn to perform these tasks

4. Compared results of these NNs with zero-shot-prompted LLMs and with fine-tuned LLMs.

**Strengths:**

The paper is fairly well-written and the experiments seem sound.

**Weaknesses:**

The paper lacks motivation for this research – there is no discussion as to why these contributions are interesting or useful. How could readers actually use these results / insights in their own work?

Proportional analogies using fixed-length Boolean feature vectors is a rather narrow notion of "analogy" in general.  How does this notion relate to the use of analogy and analogical reasoning in the real world?

The paper contains lots of formalization, but it is not clear that all of it is useful.  The paper needs more intuitive discussion of the ideas behind the formalizations (e.g., what makes an analogy “valid” or “invalid”, and why is what’s given a useful definition of this?)

**Questions:**

The paper gives a formal definition of "valid" for these proportional analogies -- can you explain intuitively what makes such an analogy "valid"?  It would help to give a concrete real-world example of a valid and invalid proportional analogy.

Formula 1: Explain what backslash symbol means in set theory

Figure 1: Explain why this analogy is considered valid.  Also give invalid example and explain why it is invalid.

Formula 6: "the following quadruplet of Boolean vectors corresponds to the valid proportional analogy of Figure 1.  Explain this representation -- how does this correspond to the figure?

For the validation task, what is the random guess baseline, given your test data split between valid and invalid?

For the generation task, there are many possible correct answers, right? what is the random guess baseline?

Table 1 is never referenced in the text.

In the caption for Table 1, specify which task this table corresponds to.

In Table 1 and Table 2, are there any error bars that could be put on these accuracy values?  E.g., using cross-validation?

For the zero-shot classification prompt given to Llama-3.1-8B-Instruct, it asks the model to indicate whether the given four images form a valid analogy".  Why would you expect that the model could know what is meant here by "valid analogy"?  I wouldn't know what you meant, just given that prompt.  Doesn't it need more information on what that means (information that is given to the fine-tuned models via the training examples)?

---

### Official Review · Reviewer_FyvQ · 2025-11-01

**Soundness:** 3
**Presentation:** 2
**Contribution:** 1
**Rating:** 2
**Confidence:** 4

**Summary:**

The paper studies analogical reasoning task for analogies of the form ((a, b), (x, y)) where x is to y as a is to b. In the paper a, b, x, y are boolean vectors. The paper presents a theoretical result on the efficient PAC learnability of these sorts of analogies. Exploring this idea experimentally, the paper builds a dataset of programatically generated analogies of this form to train and evaluate both simple NNs and LLMs. Their findings show that simple NNs learn better than LLM for both verifications and generation.

**Strengths:**

1. The paper presents a clear experiment where simple neural networks are trained on analogy data and shown to perform very well at verification and generation for data from similar distribution.

2. The paper presents a theoretical results that shows simple analogies in the boolean form are efficiently PAC learnable.

**Weaknesses:**

I do not feel the contributions of the paper are sufficient enough to warrant acceptance. While the paper demonstrates a theoretical result and clear experiments, the experiments are limited in scope and do not go beyond demonstrating the simplest implementation of the theoretical result.

1. The comparison with LLMs is not very useful. LLMs are not trained to model analogies in the representation presented in this paper. It is not clear what point the paper is trying to make with this comparison.

2. It is not clear from the paper what the significance of showing that ANNs can learn simple boolean values analogies is? Does this result have consequences towards learning complex functions beyond those results already present in learning theory literature?

3. The generalisation experiment is very limited to only studying generalisation to new entities. Does generalisation also extend to size of grid / number of shapes?

**Questions:**

1. What is the rationale for presenting boolean vectors to LLMs. Do they do better if the representation of the analogies is better suited, for example names of the shapes along with their position? Since the analogies are visual, presenting the images to VLMs maybe a better way to evaluate large pre-trained models.

---

### Official Review · Reviewer_AX76 · 2025-11-01

**Soundness:** 2
**Presentation:** 3
**Contribution:** 1
**Rating:** 2
**Confidence:** 5

**Summary:**

This paper investigates the learnability of simple proportional analogies, and compares the performance of small neural networks trained from scratch vs. LLMs in both the zero-shot and fine-tuning setting. The results show that simple neural networks can learn to verify and complete simple proportional analogies given abstract Boolean vectors as input.

**Strengths:**

- The paper clearly describes the datasets that were generated and the experiments that were performed.
- The experimental results clearly demonstrate that simple ANNs can learn to validate and complete simple proportional analogies given the right type of input representations.
- LLMs are evaluated in both the zero-shot and fine-tuned settings.
- The results are grounded in explicit theory and an accompanying proof is provided.

**Weaknesses:**

- The domain of analogy problems that's investigated in this work is extremely simple. Proportional analogy plays a relatively small role in the cognitive literature on analogical reasoning precisely because it is so simple. Proportional analogies do not involve many of the processes that make human analogical reasoning interesting and complex, including analogical mapping [1] or schema induction [2]. Moreover, the type of proportional analogies investigated in this work only involve simple synthetic relations that can be easily represented via Boolean vectors, which excludes a major part of the complexity of real-world proportional analogies, e.g. those involving real-world verbal concepts. Thus, the dataset used in this work arguably does not evaluate the majority of processes that make analogical reasoning interesting.
- Human analogical reasoning does not involve task-specific training. It is not clear what can be learned by directly training models on a specific analogy task and then testing them in-distribution. This is very far removed from the flexibility and generalizability that defines analogical reasoning.
- It is not surprising that neural networks can learn to perform this task given sufficient training, especially given the simplicity of the task and the simple synthetic nature of the inputs.
- Only relatively small LLMs are evaluated, so it is not surprising that they perform poorly in the zero-shot setting.
- LLM performance is comparable when they receive the same direct training as the other ANN model, which undermines the claimed superiority of the smaller neural network models.
- The paper emphasizes the relative efficiency of the smaller ANNs. However, as previous work has shown, LLMs are also capable of solving a much wider range of more complex and naturalistic analogy tasks, including many of those that have been commonly studied in the cognitive literature on analogical reasoning. The task studied in this paper is so simple that it would be trivial to write a simple program to solve it. Therefore, it is not clear why it is useful to have a more parameter-efficient neural network solution, especially when it requires task-specific training.

[1] Gentner, D. (1983). Structure-mapping: A theoretical framework for analogy. Cognitive science, 7(2), 155-170.

[2] Gick, M. L., & Holyoak, K. J. (1983). Schema induction and analogical transfer. Cognitive psychology, 15(1), 1-38.

**Questions:**

- How do larger and more performant LLMs perform on this task in the zero-shot setting?
- Models are evaluated in a moderately challenging OOD setting in which a specific shape or color is omitted from training and then included in testing. How are these results affected by the degree of OOD generalization that's required? If more shapes or colors are withheld from training, or more features are varied between training and test, generalization performance will presumably eventually degrade, but how does this more challenging OOD performance compare between the smaller ANNs and the fine-tuned LLMs? The current OOD essentially reflect a ceiling effect. The OOD task is too easy and so all models show accuracy close to 100%, making the comparison not very meaningful.
- Can some of the more complex aspects of analogical reasoning, such as analogical mapping or schema induction, be formalized within the framework studied in this paper?
- Given that a simple program could be written to solve the problems studied in this paper, why is it important to have parameter-efficient neural networks that can solve these problems?

---

### Note · Authors · 2025-11-20

I have read and agree with the venue's withdrawal policy on behalf of myself and my co-authors.